# Structural Insights and Catalytic Mechanism of 3-Hydroxybutyryl-CoA Dehydrogenase from *Faecalibacterium Prausnitzii* A2-165

**DOI:** 10.3390/ijms251910711

**Published:** 2024-10-05

**Authors:** Jaewon Yang, Hyung Jin Jeon, Seonha Park, Junga Park, Seonhye Jang, Byeongmin Shin, Kyuhyeon Bang, Hye-Jin Kim Hawkes, Sungha Park, Sulhee Kim, Kwang Yeon Hwang

**Affiliations:** 1Department of Biotechnology, College of Life Sciences and Biotechnology, Korea University, Seoul 02841, Republic of Korea; dkdlemf1475@gmail.com (J.Y.); intiin@hanmail.net (H.J.J.); psh3810@korea.ac.kr (S.P.); jungap@naver.com (J.P.); dbr04198@naver.com (S.J.); kingboom2@naver.com (B.S.); qptmh360@naver.com (K.B.); sulhee@korea.ac.kr (S.K.); 2Center for Creative Convergence Education, Hanyang University, Seoul 04763, Republic of Korea; hjkhawkes@hanyang.ac.kr; 3Department of Bioengineering, Incheon JEI University, Incheon 21987, Republic of Korea; m00855@jeiu.ac.kr; 4Korea BioDefense Research Institute, Korea University, Seoul 02841, Republic of Korea

**Keywords:** atopic dermatitis, 3-hydroxybutyryl-CoA dehydrogenase, serial femtosecond crystallography, acetoacetyl-CoA, open–closed conformation

## Abstract

Atopic dermatitis (AD) is characterized by a T-helper cell type 2 (Th2) inflammatory response leading to skin damage with erythema and edema. Comparative fecal sample analysis has uncovered a strong correlation between AD and *Faecalibacterium prausnitzii* strain A2-165, specifically associated with butyrate production. Therefore, understanding the functional mechanisms of crucial enzymes in the butyrate pathway, such as 3-hydroxybutyryl-CoA dehydrogenase of A2-165 (A2HBD), is imperative. Here, we have successfully elucidated the three-dimensional structure of A2HBD in complex with acetoacetyl-CoA and NAD^+^ at a resolution of 2.2Å using the PAL-11C beamline (third generation). Additionally, X-ray data of A2HBD in complex with acetoacetyl-CoA at a resolution of 1.9 Å were collected at PAL-XFEL (fourth generation) utilizing Serial Femtosecond Crystallography (SFX). The monomeric structure of A2HBD consists of two domains, N-terminal and C-terminal, with cofactor binding occurring at the N-terminal domain, while the C-terminal domain facilitates dimerization. Our findings elucidate the binding mode of NAD^+^ to A2HBD. Upon acetoacetyl-CoA binding, the crystal structure revealed a significant conformational change in the Clamp-roof domain (root-mean-square deviation of 2.202 Å). Notably, residue R143 plays a critical role in capturing the adenine phosphate ring, underlining its significance in substrate recognition and catalytic activity. The binding mode of acetoacetyl-CoA was also clarified, indicating its lower stability compared to NAD^+^. Furthermore, the conformational change of hydrophobic residues near the catalytic cavity upon substrate binding resulted in cavity shrinkage from an open to closed conformation. This study confirms the conformational changes of catalytic triads involved in the catalytic reaction and presents a proposed mechanism for substrate reduction based on structural observations.

## 1. Introduction

Atopic dermatitis (AD), also referred to as atopic eczema, is a persistent inflammatory cutaneous disorder typified by pruritic, erythematous, and desquamating skin lesions, and frequently confined to the anatomical regions characterized by flexion [1]. According to the Korean National Health Insurance Service (NHIS), the highest prevalence of AD (26.5%) was observed in children aged 12 to 23 months in 2008, decreasing with age. In addition, the overall prevalence in children under 18 slightly increased from 4.0% in 2003 to 4.5% in 2018, with the prevalence in those aged 6 to 18 rising from 1.9% to 3.1% [2,3]. Clinical manifestations of AD tend to ameliorate with maturation in a subset of affected children. However, approximately half of these children may progress to develop allergic asthma, while approximately two-thirds are at risk of developing allergic rhinitis [3]. In addition to its prevalence in Korea, AD has shown an elevated incidence in other developed nations [4,5]. While environmental factors and genetic predisposition have been considered as primary etiological factors, the rapidity of this phenomenon surpasses any plausible genetic shifts, thereby implicating alterations in the environment associated with Westernized lifestyles as the likely contributing elements [6]. A recent study has shed light on the association between a Westernized diet, characterized by reduced fiber consumption, and the occurrence of distinct microbiome dysbiosis. The intestinal microbiota is extremely plastic in response to diet and environmental factors and, at the same time, governs many aspects of the immune function via short-chain fatty acids (SCFAs) throughout the body [7]. SCFAs are metabolic byproducts originating from the microbiota, comprising a carboxylic acid segment linked to a hydrocarbon tail. SCFAs are mainly produced by microbes in the gut and by skin-commensal bacteria at low concentrations [8]. The significance of SCFAs in AD has been substantiated through research spanning several decades. The gut–skin axis serves as the foundation for systemic allergen sensitization primarily facilitated by the production of SCFAs, notably butyrate. Butyrate enhances skin barrier integrity by modulating the mitochondrial metabolism of epidermal keratinocytes and influencing the synthesis of crucial structural constituents [9,10]. Furthermore, butyrate exerts inhibitory effects on *Staphylococcus aureus*, a microorganism implicated in the upregulation of Th2 (T helper cell 2) cytokines in individuals with AD. This inhibition exacerbates dermal IL-33 expression and skin inflammation via the inhibition of histone deacetylase3 (HDAC3) [11]. Hence, the production of butyrate by the intestinal microbiota exerts a protective influence on AD, contributing to the mitigation of the condition. By the investigation conducted by Han Song [10], a comparative analysis of fecal samples from individuals afflicted with AD and those without the condition revealed a noteworthy reduction in the levels of butyrate, a substance known for its anti-inflammatory properties, in AD patients, and distinctive dysbiosis of microbiota. Pan-genome sequencing of five *Faecalibacterium prausnitzii* strains (A2-165, M21/2, L2-6, S3L/3, and KLE1255) revealed distinctive genetic elements that are responsible for the utilization of significant mucin constituents, such as GalNAc and L-fucose, as well as various nutrients typically released from compromised gut epithelial cells [10]. Notably, the strain L2-6 exhibited a higher count of mapped AD-associated reads compared to non-AD reads, whereas the A2-165 strain displayed the converse pattern [10,12]. Furthermore, the investigation unveiled that dysbiosis within the *F. prausnitzii* species, linked to AD, could potentially lead to the suppression of high butyrate-producing subspecies, notably the A2-165 bacterial type. This was substantiated through a comparative analysis of butyryl CoA: acetate CoA-transferase gene promoter activity among various *F. prausnitzii* strains, suggesting a plausible mechanism resulting in an overall decrease in butyrate production [10,12]. Consequently, comprehending the function of key enzymes such as thiolase, 3-hydroxybutyryl-CoA dehydrogenase, butyryl-CoA dehydrogenase, and butyryl-CoA:acetate CoA transferase in the butyrate pathway is highly significant. Although the structures of these enzymes contributing to the production of various butyrates have been reported [13,14], there have been no comparative studies investigating a difference in production despite the same subspecies. To understand the differences in butyrate production between the L2-6 and A2-165 subspecies, we will aim to clarify the architectures of key enzymes from both subspecies. The A2HBD has elevated expression in *F. prausnitzii* A2-165 and is crucial for the transformation of acetoacetyl-CoA (Appendix A) to 3-hydroxybutyryl-CoA in the butyrate cycle. We describe the whole structure of A2HBD, including the substrate and co-factor binding modes, as derived from three crystal structures (Figure 1a and Figure 2a,b). Additionally, informed by these structural findings, we propose a putative molecular mechanism for the A2HBD enzyme. In the tertiary-A2HBD structure, the electron density of acetoacetyl-CoA and NAD^+^ were observed in only one protomer. We propose that this enzyme exhibits cooperativity upon substrate binding, resulting in a conformational change of the cleft. Furthermore, our biochemical findings demonstrated that the binding affinity of acetoacetyl-CoA is less stable than that of NAD^+^.

## 2. Results

### 2.1. Overall Structures of Apo-A2HBD Complex with NAD^+^ and Acetoacetyl-CoA

The crystal structure of apo-A2HBD was determined at 2.55 Å resolution. The apo form of A2HBD belonged to P2_1_ space group with unit cell parameters a = 87.660 Å, b = 126.611 Å, c = 108.071 Å and α = 90 Å, β = 110.208, γ = 90 Å. The apo-A2HBD protein assembles into an asymmetrical hexameric configuration (Figure 1b), wherein all six subunits adopt nearly identical conformations, displaying root-mean-square deviations (RMSD) for the Cα atoms within the range of 0.2 to 0.378 Å (Figure 1a). In the asymmetric unit of crystals, it contains three dimers with three two-fold axes (Figure 1a, Figure 2a and Appendix A). The crystal structure of the tertiary complex of A2HBD was solved at 2.2 Å resolution (Figure 2a). The tertiary complex belonged to P3_2_12 space group with unit cell parameters a = 90 Å, b = 90 Å, c = 212 Å and α = 90 Å, β = 90, γ = 120 Å. Notably, the tertiary-A2HBD protein manifested as a trimer, wherein only one subunit harbored acetoacetyl-CoA and NAD^+^ (Appendix A, Figure 2a and Figure 3a). When the merged hexamer structure (trimer and the crystallographic symmetric unit of the trimer) was superimposed onto the apo-hexamer, the overall structural congruence was strikingly high, with an RMSD of 0.375Å. Furthermore, compared to the dimerization interface observed between the A and C subunits, the B and B’ subunits exhibited a similar dimerization interface (Figure 4). The A2HBD protein exists as a hexamer in solution (Figure 1b). In the tertiary-A2HBD structure, the electron densities of acetoacetyl-CoA and NAD^+^ were observed in only one protomer. The exact reason for the complex formation involving only one subunit remains unclear. We propose that, upon substrate binding, the cleft formed between the A, C, and B, B′ subunits contracts by approximately 11 degrees (Figure 2b,c). This hypothesis suggests that this contraction of the dimeric cleft may trigger conformational changes within the hexamer structure, which in turn could affect the solvent accessibility of the other dimer. Such structural alterations might influence the enzyme’s substrate preference.

### 2.2. Monomer of A2HBD

The composition of A2HBD’s monomer structure displayed a two-domain topology, featuring distinct N-terminal and C-terminal domains (Figure 3a). The monomer structure consists of eleven α-helices and ten β-strands arranged sequentially, extending from the N-terminus to the C-terminus. Within the N-terminal domain (residues 1–182), a cohesive core structure is formed by the arrangement of seven β-strands, flanked by α-helical regions, among a total of ten β-strands. This core structure shows a typical Rossmann fold pattern, commonly seen in many NAD (P)-dependent dehydrogenases. Similar to a standard Rossmann fold, the first five strands (β1 to β5) in the sheet are arranged side by side, oriented in the same direction. The last two β-strands also exhibit a parallel configuration; however, they extend in the opposite direction compared to the first six strands (Figure 3a). A2HBD also features another unique structural sub-domain known as a helix-turn-helix motif (*Clamp-roof domain*, as herein referenced). This motif (α2, α2- α3, α3) extends from the central Rossmann-fold structure, specifically connecting the β2 and β3 strands. This motif contains several charged and hydrophobic amino acids, including lysine and alanine. Some of these may play a role in interacting with the adenine diphosphate moiety of the acetoacetyl-CoA substrate. Predominantly composed of α-helices, the C-terminal domain (residues 183–288) plays a pivotal role in subunit dimerization (Figure 4). It generates a deep cleft, mainly by α9 and α10 (*Clamp-base domain*), together with the Clamp-roof domain, thereby stabilizing the adenine phosphate moiety of acetoacetyl-CoA (Figure 2b and Figure 3a,b).

Within a single asymmetric unit, two A2HBD protomers—specifically the A and C subunits—engage in mutual contact facilitated by their C-terminal domains, forming a dimeric configuration. The dimerization process is predominantly driven by hydrophobic interactions (Figure 4). The primary contributors to these essential hydrophobic interactions include the α9 and α10 helices, as well as the β7 and β8 strands. It is worth noting that the hydrophobic residues responsible for mediating this dimerization process remain remarkably conserved among homologous HBD proteins from various species. These residues encompass Val183, Val184, Ile187, Leu188, Ile189, Pro190, Met191, Ile192, Phe197, Ile198, Met200, Val203, Met213, and Leu215. Additionally, the unique antiparallel arrangement of the α9 and α10 helices is further stabilized by the formation of two pairs of salt bridges. Specifically, these salt bridges are formed through interactions between Arg187 and Glu195, as well as Glu202 from one subunit and their counterparts from the other subunit. This Arg/Glu interaction also exhibits high conservation among HBD homologs from diverse species (Figure 4 and Appendix Ab).

### 2.3. NAD^+^ Binding Site of A2HBD

To ascertain the binding mode of NAD^+^, we determined the 2.55Å resolution crystal structure of A2HBD in complex with NAD^+^ cofactor (Figure 3a–c). In the structural analysis of the NAD^+^ complex derived from crystallography within the P2_1_ space group, it was observed that all subunits comprising the hexameric assembly were found to bind to NAD^+^. By superposing the NAD-bound form of A2HBD with the apo-enzyme, we observed that the monomer structure of the NAD complex is similar to the apo-enzyme structure, displaying a root-mean-square deviation of 0.543Å. Like other NAD-dependent dehydrogenase enzymes, the NAD^+^ cofactor binds to the Rossmann fold domain without forming significant interactions with the C-terminal domain (Appendix A). Observing the binding mode of the NAD cofactor, adenine moiety of NAD cofactor is somewhat exposed to the surface of the enzyme, and the ribose and nicotinamide portion fitted into the deep cleft facing the acetyl group of the substrate. The phosphate moiety of the NAD cofactor resides within the G-x-G-x-x-G motif, composed of Gly7-Ala8-Gly9-Thr10-Met11-Gly12, situated between the initial β strand and the first α a helix within the Rossmann fold domain (Appendix A). The NAD cofactor is captured by the enzyme’s Thr10 and Met11 main-chain nitrogen atoms, establishing hydrogen bonds with the hydroxyl group of the phosphate moiety. Hydrogen bonding interactions involving Asp31, Glu90, Lys95, Asn115, and Ser117, along with hydrophobic interactions with Met11 and Phe138, contribute to the stabilization of the NAD cofactor’s two ribose rings and nicotinamide moiety. The hydrophobic pit accommodates the adenine moiety, fostering interactions with hydrophobic residues such as Ile6, Ile32, Ala88, Phe,89, Val94, and Thr98; thus contributing to the stabilization of the adenine phosphate (Figure 3c and Appendix A).

### 2.4. Acetoacetyl-CoA Binding Sites of A2HBD

The tertiary complex of HuHAD with NAD^+^ and acetoacetyl-CoA demonstrates a notable conformational change upon substrate binding [15]. Specifically, the NAD^+^ -binding domain (N-terminal domain) undergoes an inward rotational motion towards the C-terminal domain, facilitating a robust substrate binding interaction [15]. To ascertain if A2HBD exhibits a similar conformational change, we determined the crystal structure of A2HBD in the complex with cofactor and substrate at 2.2 Å (Figure 2a, Figure 3a and Figure 5a). The acetoacetyl-CoA substrate is located between the clamp-roof domain (moving-helix) and clamp-base domain (Figure 5a). Next, the conformational differences of A2HBD with the structures of NAD-complex and tertiary complex were compared. When superposed to the NAD^+^ -bound form of the enzyme with the tertiary structure, the overall conformation has been changed, displaying an RMSD of 2.202 Å (Figure 5a,b). Interestingly, overt dissimilarity in structure was conspicuously evident within the N-terminal domain, whereas the C-terminal domain displayed more subtle and discrete conformational alterations. The most notable aspect of this structural change is the movement of the clamp-roof domain, which shifts approximately 6.02 Å closer to the acetoacetyl-CoA compared to its position in the NAD complex, with a rotation of 11 degrees (Figure 5b). This may suggest that when binding of the cofactor, there is no requirement for domain displacement; rather, the transition from an open to a closed conformation plays a pivotal role in facilitating substrate binding (Figure 5c). This hypothesis gains support from the observation that the nicotinamide ring of NAD^+^ undergoes an inward movement towards the catalytic cavity during the oxidoreductase reaction with the acetoacetyl moiety of the substrate. The transition from the open to the closed conformation also triggers secondary structural changes, resulting in the formation of the catalytic cavity that contains the catalytic residues. In the open conformation, the cavity′s diameter measures approximately 7.5 Å, allowing for ample substrate entry (Figure 5d,e). In contrast, in the closed conformation, this diameter reduces to 5.2 Å, a contraction of 55%. Following substrate binding, the main chain of Asn139 established hydrogen bonds with the substrate, allowing the side chain of Asn139 to align in a manner conducive to Ar-HN interactions (weakly polar interactions between aromatic rings of amino acids and hydrogens of backbone amides) with the aromatic ring of Phe230 (Figure 5d,e). Furthermore, the side chain of Phe230 is closely packed with hydrophobic residues such as Ile231 and Leu227, among others, forming a hydrophobic core near the active site (Figure 5e). In the open conformation, these interactions were disrupted due to alterations in the positioning, geometry, and distances of the residues surrounding Phe230 [16]. It is postulated that the catalytic cavity, shaped by interactions involving Phe230, potentially plays a role in orchestrating the arrangement of catalytic residues, thereby facilitating the occurrence of oxidoreductase activity.

The tertiary structure analysis also elucidated the substrate binding mode of A2HBD. Due to the limited resolution of the electron density map in the adenine moiety of acetoacetyl-CoA obtained from the tertiary structure analysis, a detailed examination of the binding mode between the adenine moiety of the substrate and the enzyme was conducted using fourth-generation PAL-XFEL diffraction data (Figure 6g and Appendix A). The structures formed almost the same conformation with the RMSD of 0.19Å. Compared to the NAD-binding mode, the acetoacetyl-CoA substrate forms fewer hydrogen bonds with the enzyme (Figure 6e,f), in contrast to the stronger interactions observed with the NAD cofactor, as illustrated in Figure 3d. Therefore, numerous crystallography datasets obtained from the A2HBD, NAD, and acetoacetyl-CoA co-crystals consistently revealed an unclear electron density map for the adenosine phosphate within the substrate (Appendix A). This suggests a limited level of stabilization for the adenosine ring by the enzyme. This moiety is exposed to the surface, having no binding pocket for the interaction (Figure 5). Instead, Lys54 (NZ) and Lys56 (NZ), located in the clamp-roof domain, form hydrogen bonds with ribose (O48) and O53 of 3′phosphate, respectively. Arg143 (NH1) forms a hydrogen bond with O30 of the substrate within a 3.00 Å distance (Figure 6e,f). Interestingly, in the tertiary structure, rotations of Arg143 toward the adenine ring were observed compared to the NAD complex, allowing the acetoacetyl-CoA to be stabilized through a cation-π interaction. This stabilization occurs despite the limited involvement of other residues in securing substrate binding (Figure 6e,f). The guanidinium moiety is oriented towards the aromatic ring of the adenine moiety in a stacked (parallel) configuration, with a proximate distance (below 4 Å) between the guanidinium and adenine planes. This arrangement suggests the formation of a cation-π interaction that contributes to the substrate′s stabilization [17]. Asn139 mainchain oxygen is positioned to hydrogen bond to the Pantothenic group. The acetoacetyl group is located in hydrogen bonding distance with the sidechains of Ser117 (OB), His136 (NE2), and Asn186 (ND2). These residues interacting with the acetoacetyl moiety are highly conserved (Appendix A). The hydroxyl group of Ser117 establishes an interaction with the NAD cofactor, thus placing this residue in a bridge with both the substrate and the cofactor. The sidechain of Asn186 located in the C-terminal domain also forms hydrogen bonding with the acetoacetyl moiety of the substrate. Biochemical assays and mutagenesis studies have not yet been undertaken to clarify the primary functions of these residues. Nevertheless, the observations made in this study offer valuable structural insights into the underlying mechanism governing the enzyme’s activity.

## 3. Discussion

The structural analysis of the NAD^+^ and tertiary complex structures reveals that the catalytic residues, specifically Ser117, His136, and Glu148, are positioned near the active site. This domain encompasses both the acetyl group of the substrate and the amide group of the NAD^+^ nicotinamide, suggesting their essential roles in the catalytic process (Appendix A and Figure 7a). These residues are part of the conserved sequence, HXFXPXXXMXLXE, which shows strong conservation across the HBD families, as noted in previous studies [18,19,20]. The molecular mechanism underlying HBD catalysis involving this conserved catalytic domain has been thoroughly investigated in prior research [17,18,19,20]. The validity of these mechanisms has been substantiated through the analysis of the crystal structure of the A2HBD tertiary complex. The structures revealed that Ser117, His136, and Glu148 constitute the catalytic triad, engaged in hydrogen bonding interactions within the NAD^+^ complex (Appendix A). When comparing the catalytic triad′s conformational changes from the open to closed conformations (Figure 5c and Figure 7a), it becomes evident that in the NAD^+^ complex, characterized by an open conformation, the catalytic triad resides within a hydrogen bonding distance from each other, thereby contributing to the stabilization of these critical residues (Appendix A). In the crystallization solution, assessed at a pH range of 5.3 to 5.6, it was observed that the imidazole ring of His136 exists in a protonated state, while the side chain of Glu148 remains negatively charged. This finding indicates that the protonated state of His136 is stabilized by the negative charge present on Glu148, which is referred to as the His/Glu pair, a phenomenon also supported by observations in the case of human 3-hydroxyacyl-CoA dehydrogenase [21]. After substrate binding, NAD^+^ undergoes an inward movement, positioning the nicotinamide moiety within the active site. Simultaneously, the acetoacetyl group of the substrate is situated adjacent to the nicotinamide moiety within the active site. These specific placements result in Ser117 being positioned beyond the hydrogen bonding distance with His136, measured at approximately 4.9 Å.

However, the His136 and Glu148 maintain hydrogen bonding interactions, with His136 notably forming a bond with the O3 atom of the substrate at a distance of approximately 2.9 Å (Figure 6b and Figure 7a). The disruption of hydrogen bonding between Ser117 and His136 in a substrate-dependent manner suggests a potential role for His136 as a general acid/base catalyst and sheds light on the mechanism of nucleophilic reactivity within the enzyme (Appendix A). Under the modulation of the catalytic triad through the open-closed conformational transition, we could confirm and propose a conserved mechanism for acetoacetyl-CoA-catalyzed reactions in A2HBD, as shown in Figure 7a. Upon substrate binding, acetoacetyl-CoA is stabilized in the closed conformation through interactions with the conserved active site residues (Figure 6e and Appendix A). The positive charge carried by the protonated imidazole ring of His136 plays a crucial role in stabilizing the partial negative charge on the C3 oxygen (O3) within the transition state of the substrate. Additionally, it aids in polarizing the carbonyl moiety of the substrate. Following this polarization, the carbonyl group develops an electron-deficient carbon, facilitating the transfer of a hydride ion from NADH to the C3 carbon. Subsequently, His136 functions as a proton donor, leading to the reduction of acetoacetyl-CoA to 3-hydroxybutyryl-CoA. This deprotonation event disrupts the hydrogen bonding interaction with Ser117 (Appendix A and Figure 7a). The binding affinity of acetoacetyl-CoA to A2HBD is 23.9 μM while that of the NAD^+^ is 346.6 μM (Figure 7b). In the case of NAD^+^, its binding affinity for A2HBD (31.5 μM) is stronger when the substrate (acetoacetyl-CoA) is present, compared to when it is alone (Figure 7b). We may suggest that this is related to the formation of the tertiary complex in only one monomer within the trimer, which may help explain the cooperative nature of this enzyme (Figure 2a). While the structural data from A2HBD supports the proposed catalytic mechanism, further confirmation of the roles played by the catalytic residues necessitates site-directed mutagenesis and enzymatic assays in future investigations.

## 4. Materials and Methods

### 4.1. Cloning and Protein Expression

The A2-165 BHBD coding gene (Met1-Leu290) with a calculated molecular weight of 31.1 kDa, underwent enzymatic amplification via polymerase chain reaction (PCR) utilizing chromosomal DNA isolated from *F. praunsnitzii* as the template. Subsequently, the resulting PCR product of the A2-165 strain (DSM 17677) was integrated into a pET21a^+^ vector (Novagen, Sigma-Aldrich, Burlington, MA, USA) containing a 6x histidine tag at the C-terminus. Amplification of gene fragments was achieved employing primers 5′-**ctagctagc**atgaagatcggtgttatc-3′ (A2HBD_forward) and 5′-**ccgctcgag**ctggtcaacaggggtc-3′ (A2HBD_reverse), resulting in the amplification of the full-length 867bp A2-165 BHBD gene. Notably, the bold sequence section in the primer design corresponds to the incorporation of *NheI* and *XhoI* restriction enzyme recognition sites at the 5′-end of the forward and reverse primers, respectively. The constructed expression vector was then introduced into *Escherichia coli* BL21 (DE3) for protein expression. The transformed cells were cultured in 4L of LB medium containing ampicillin (50 mg/mL) at 37 °C, until reaching an absorbance 0.6~0.8 at OD_600_. Protein production was induced by 1.0mM Isopropyl β-D-1-thiogalactopyranoside (IPTG), and then the cells were incubated for 18 h at 18 °C. The grown cells were harvested by centrifugation at 1600× *g* at 277.15 K for 1 h. The cell pellets were resuspended in lysis buffer (20 mM Tris-HCl at pH 8.0 and 150 mM NaCl) and then disrupted by ultrasonication.

### 4.2. Protein Production and Crystallization

The cell debris was removed by centrifugation at 13,000 rpm for 60 min. The supernatant was filtered through a 0.45 μm polyvinylidene difluoride (PVDF) membrane filter (Merck Millipore, Seoul, Republic of Korea) and was bound to a 5 mL His Trap HP column (GE Healthcare, Seoul, Republic of Korea). After washing with His binding buffer (20 mM Tris-HCl at pH 8.0 and 150 mM NaCl), the bound protein was then eluted with His binding buffer containing 500 mM imidazole in a linear gradient. To eliminate trace contaminants and facilitate crystallization, the protein sample was subjected to size-exclusion chromatography using a HiLoad 16/600 Superdex 75 prep-grade column (GE Healthcare). The column was pre-equilibrated with a final buffer solution consisting of 20 mM Tris-HCl at pH 8.0 and 150 mM NaCl. Following chromatography, the purified protein, which exhibited an approximate purity of 95%, as determined by SDS-PAGE analysis, was concentrated to a final concentration of 53.3 mg/mL. This concentration step was achieved using an Amicon filter (Merck, Seoul, Republic of Korea) with the final buffer as the processing medium.

The initial screening for the crystallization was performed by the sitting-drop vapor diffusion method using the commercial crystallization screening kit (Hampton Research) in MRC 2-well Crystallization Plates (Hampton Research, Seoul, Republic of Korea) at 293 K. MCSG crystal screening solution (Anatrace, Maumee, OH, USA) was used as a screening solution. A volume of 60 μL of the screening solution was dispensed into each reservoir well, and the screening process was conducted by combining 0.5 μL of the screening solution with an equivalent amount of protein sample. After a 14-day incubation period at a temperature of 293 K, we successfully generated crystals suitable for diffraction experiments. These crystals were formed through co-crystallization of the substrate, cofactor, and tertiary complex using the same method and screening conditions. Notably, the crystallization process was supplemented with 5 mM each of NAD^+^ and acetoacetyl-CoA. The crystal solution conditions were further optimized in a hanging-drop vapor diffusion method adjusting the pH, salt, and precipitant concentration. 1 μL of 20 mg/mL protein solution was mixed with precipitation solution at a ratio of 1:1 in 24 well VDX plates (Hampton Research) at 293 K. The method employed for creating the complex crystal was identical, except for the omission of 5 mM each of NAD^+^ and acetoacetyl-CoA. After going through several steps for optimization, the best quality of crystals and the maximum size of macro-crystals appeared after 14 days in a solution condition of 0.23 M Ammonium citrate dibasic (pH 5.0) and 1 5% PEG3350 for the apo- and tertiary complex. For NAD^+^ complex, the solution condition of 0.25 M Lithium Sulfate, 0.1 M Tris-HCl, and 1.28 M Ammonium Sulfate produced the best-quality crystals [22].

In pursuit of generating a substantial quantity of micro-crystals for serial femtosecond crystallography (SFX) experiments, which necessitate the acquisition of diffraction data from a multitude of micro-crystals possessing random orientations [23], a crystallization process was initiated via batch-crystallization [24]. Before crystallization, 20 mg/mL of A2HBD protein solution was incubated with 5 mM acetoacetyl-CoA and 5mM NAD^+^ cofactor for 30 min. In the initial batch experiment, we amalgamated 10 μL of protein solution with a concentration of 20 mg/mL and a precipitant solution comprising 0.21 M to 0.28 M Ammonium citrate dibasic (pH 5.0) along with 15% PEG 3350, utilizing ratios of 1:1, 1:2, and 1:3, respectively. The resulting mixture was subjected to gentle mixing within Eppendorf tubes (EP tubes) using a pipette, followed by incubation in sealed EP tubes at 293 K to prevent evaporation. Notably, during this step, meticulous care was taken to ensure a slow addition of the precipitant solution to the protein, with a duration of 2–5 s, to avert the formation of localized high concentrations of precipitants [24]. Following the 5~7 days incubation period, crystals with dimensions less than 0.2 mm were successfully obtained in the mixture solution of a ratio of 1:3.

### 4.3. Data Collection for Apo, NAD^+^ Complex and Tertiary Comple

Before data collection at 100 K, the crystals were transferred to cryoprotectant solution containing 0.23 M ammonium citrate dibasic (pH 5.0), 15% PEG3350 and 25% glycerol. The crystals were flash-frozen in liquid nitrogen and then processed into a cryoprotectant solution containing 0.23 M ammonium citrate dibasic (pH 5.0), along with a mixture of 15% PEG3350 and 25% glycerol. For NAD^+^ co-incubated crystals, the cryoprotectant solution included 0.25 M lithium sulfate, 0.1 M Tris/HCl, 1.28 M ammonium sulfate and 25% 2-Methyl-2,4-pentanediol [22]. Additionally, 20 mM of cofactor and substrate-cofactor were introduced into the mixture. Following this, the crystals were promptly flash-frozen by immersion in liquid nitrogen. The X-ray diffraction data were collected at Pohang Accelerator Laboratory (PAL) beamline 11C. The diffraction frames were obtained on the PILATUS3 X 6 M detector using 1° rotation and 1s exposure time for 360°. The data set was indexed, integrated, and scaled with the HKL2000 suite [25].

The X-ray diffraction data of the acetoacetyl-CoA complex were collected at the Pohang Accelerator Laboratory X-ray Free Electron Laser (PAL-XFEL) facility. The fixed-sample scanning method was utilized to precisely position and transport the crystal sample to the designated X-ray interaction point [26]. For the sample delivery system, a Nylon-Mesh sample holder was employed. This holder, composed of Nylon-Mesh containing crystals enveloped by polyimide films, offered the advantage of minimal background scattering when subjected to X-rays. This characteristic allowed us to circumvent issues related to dehydration and enabled the acquisition of highly precise diffraction data for subsequent analysis [27]. Prior to mounting, the crystal solutions were pipetted to fragment the crystals into sizes below 200 µm, ensuring they could be effectively positioned on the pores of the Nylon-Mesh sample holder. The sample was then mounted inside the FT-SFX chamber equipped with helium purging environments. The diffraction images were captured using a Rayonix NX225-HS detector. Using a Cheetah software package (version 1.12), the data were indexed, integrated, and scaled [28].

### 4.4. Structure Determination

In the process of deriving a model structure for the protein crystals, the Python-based Hierarchical Environment for Integrated Xtallography (Phenix) software program (version 1.20-4459) was employed [29,30]. This involved inputting a protein sequence, a data file, and a reference model. Initially, we solved the structure through molecular replacement (MR) using the 3-hydroxybutyryl-CoA dehydrogenase from *Clostridium butylicum* (PDB ID: 4KUE) as the reference model, utilizing the Phaser module within PHENIX [29]. Subsequently, 90% of the residues were automatically built using the Autobuild module in PHENIX [30]. Further refinement and manual model building were conducted with the assistance of the Crystallographic Object-Oriented Toolkit (Coot 0.9.8.95) [31]. The model structure was fitted with an electron-density map using Coot 0.9.8.95, and final refinement was executed using the Refine module within PHENIX. Further model building and refinement were performed using REFMACS in CCP4 (version 0.9.0.003) and PHENIX (version 1.21). The final model for apo-A2HBD had R values of R_factor_ = 21.5%, R_free_ = 25.7% at 2.75 Å resolution; NAD^+^/A2HBD had R values of R_factor_ = 20.6%, R_free_ = 26.6% at 2.55 Å resolution; acetoacetyl-CoA/A2HBD (S117A) had R values of R_factor_ = 18.0%, R_free_ = 22.2% at 1.90 Å resolution; NAD^+^/acetoacetyl-CoA/A2HBD (S117A) had R values of R_factor_ = 19.9%, R_free_ = 24.4% at 2.2 Å resolution. Finally, the structural figures were generated using PyMOL 3.0 (Educational version) [32]. The statistics for the detailed data collection and structure refinement are provided in Table 1.

### 4.5. Multi-Angle Light Scattering (MALS) Analysis

Multi-angle light scattering (MALS) was conducted to determine the absolute molecular weight of the A2HBD. The AKTA chromatography system was integrated with a MALS detector TREOS II MALS (WYATT). The purified protein concentration was diluted in the final buffer (20 mM Tris-HCl at pH 8.0 and 150 mM NaCl) to 2.018 mg/mL and analyzed using SEC-MALS. Protein solution was loaded on a Superdex 200 Increase analytical column (GE Healthcare), which is pre-equilibrated in a final buffer. The buffer′s flow rate was consistently upheld at 0.3 mL/min. Before experimenting, the bovine serum albumin (BSA) was quantified as a baseline reference measurement. ASTRA software (https://wyatt.com/products/software/astra.html, accessed on 30 September 2024, Wyatt Technology, Goleta, CA, USA), provided by MALS manufacturer, was utilized to process and analyze the data. The elution profile of A2HBD exhibited a predominant peak, which accounted for 92% of the scattered mass and was estimated to be approximately 212.2 kDa with a precision of ±1.577%. Considering its monomer molecular weight of 32.4 kDa, this mass value provides the estimate for a hexameric state when A2HBD is present in the solution (Figure 1b).

### 4.6. MicroScale Thermophoresis (MST) Measurement

MST measurement was conducted to identify molecular interaction between A2HBD and substrate or cofactor. A2BHD protein was labeled fluorescently using a His-tag Labeling Kit (The Monolith His-Tag Labeling Kit RED-tris-NTA 2nd generation: Nanotemper technologies, South San Francisco, CA, USA) at 50 nM concentration. We started at 0.55 mM, and 0.2 mM, respectively. The proteins were serially diluted to 16 concentrations. 1:1 protein–protein solutions were incubated for 30 min. The measurements were conducted under conditions where the LED power and MST power were both at 40%. The affinity constant (Kd) was determined by curve fitting using MO Affinity Analysis software v2.1.3 (Nanotemper, South San Francisco, CA, USA). All measurements were conducted in triplicate (Figure 7b).

## 5. Conclusions

Some scientists have been studying the link between butyrate metabolism and atopic dermatitis (AD) [10,12]. To understand the butyrate metabolism pathway, we have determined the four crystal structures of 3-hydroxybutyryl-CoA dehydrogenase from *Faecalibacterium prausnitzii* strain A2-165 (A2HBD) in the presence of acetoacetyl-CoA and NAD^+^. This was achieved using the PAL-11C beamline (a third-generation facility) and PAL-XFEL (a fourth-generation facility) with the technique of serial femtosecond crystallography (SFX). In the tertiary-A2HBD structure, the electron density of acetoacetyl-CoA and NAD^+^ were observed in only one protomer. We propose that this enzyme exhibits cooperativity upon substrate binding, resulting in a conformational change of the cleft. Furthermore, our biochemical findings demonstrated that the binding affinity of acetoacetyl-CoA is less stable than that of NAD^+^. The Arg143 has a critical role in identifying substrates and facilitating catalytic activity. The binding mechanism of acetoacetyl-CoA was studied further, revealing that it is less stable than NAD^+^. This work validates the conformational changes of catalytic triads involved in the catalytic reaction. Furthermore, it presents a theorized mechanism for substrate reduction based on structural discoveries. Our biochemical findings demonstrated that the binding affinity of acetoacetyl-CoA is less stable than that of NAD^+^.

## Figures and Tables

**Figure 1 ijms-25-10711-f001:**
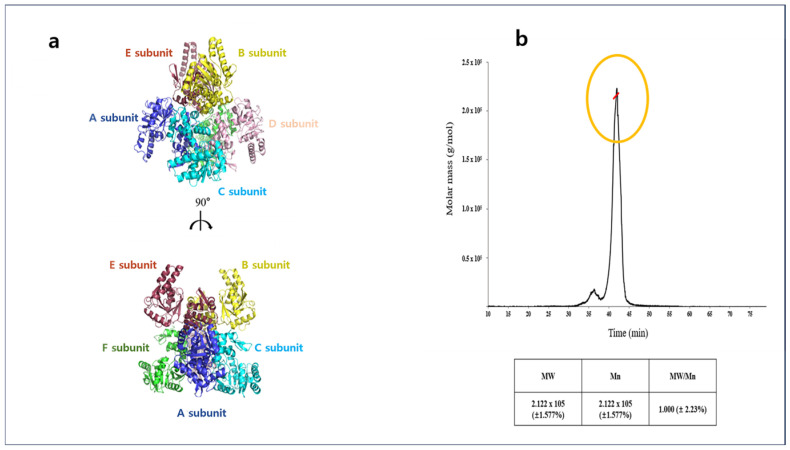
(**a**) Overall structure of apo-A2HBD: Each subunit of hexameric A2HBD is distinguished with different colors (**b**) SEC-MALS of apo-3-Hydroxybutyryl-CoA (A2HBD); sample was measured at 2.018 mg/mL concentration. The protein sample was separated on a size exclusion chromatography (SEC) column. The horizontal line marked as red in yellow circle. represents the measured molar mass. The peak (black) represents the light scattering.

**Figure 2 ijms-25-10711-f002:**
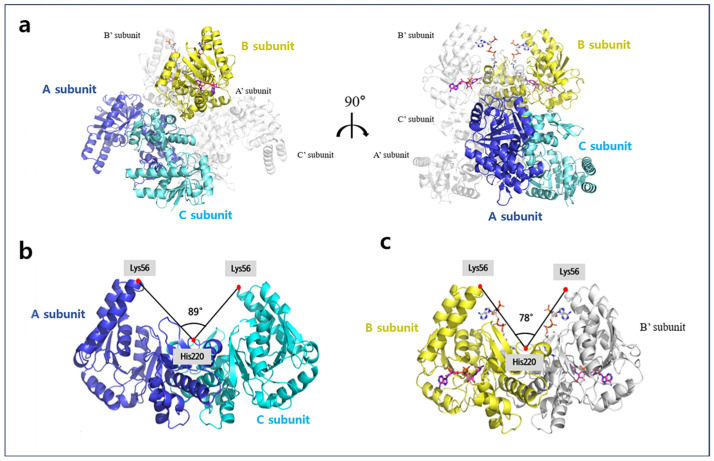
The overall structure of A2HBD in complex with NAD^+^ and acetoacetyl-CoA. (**a**) The merged hexamer of the A2HBD trimer and the symmetric subunit are shown as a cartoon. The trimeric asymmetric unit of the tertiary complex is shown as a colored cartoon. Subunits A and C are labeled as blue and cyan, respectively. The B subunit of the tertiary complex is shown in yellow. The bound ligand NAD^+^ and acetoacetyl-CoA are shown in magenta and white, respectively, in a stick model. The symmetric structure of A2HBD is shown in white. (**b**) Comparison of the cleft angle in subunit dimers. Non-bound subunits A and C dimer are shown as cartoons. Subunits A and C are colored blue and cyan, respectively. (**c**) The merged dimer of ligand-bound subunit B and its symmetric unit are shown as cartoons with yellow and white, respectively. The cleft angle is defined by Lys56 at the edge of α2 of the two subunits and His220 at the center of the dimer. The tilted cleft angle is further narrowed by the addition of ligands. Ligands of NAD^+^ and acetoacetyl-CoA are shown as stick models with magenta and white, respectively.

**Figure 3 ijms-25-10711-f003:**
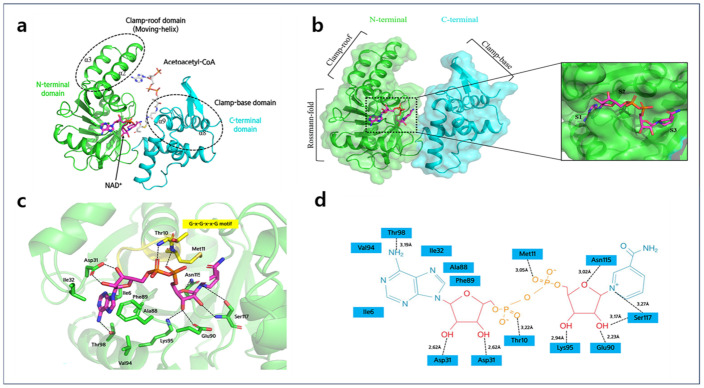
The overall structure of A2HBD in complex with NAD^+^ and acetoacetyl-CoA. (**a**) Monomer structure of A2HBD complexed with NAD^+^ and acetoacetyl-CoA. The tertiary monomeric structure is depicted in a cartoon representation, with the N-terminal domain highlighted in green and the C-terminal domain in cyan. The presence of bound NAD^+^ and acetoacetyl-CoA is illustrated using a stick model, with NAD^+^ represented in magenta and acetoacetyl-CoA in white. The regions responsible for substrate binding, depicted as the clamp-roof and clamp-base domains, are indicated by a dotted circle in black and are appropriately labeled. (**b**) Crystal structure of A2HBD in complex with NAD^+^. (left) The “open-book” view of A2HBD monomer is shown in both the cartoon and surface diagram. Three sub-domains and two domains are labeled. The stick diagram of NAD^+^ is colored magenta. (right) An enlarged view of the cofactor-binding pocket. NAD^+^ is shown as a stick and A2HBD is shown as a surface. The subsites (S1, S2, S3) of A2HBD are labeled. (**c**) The detailed interaction between NAD^+^ and A2HBD. The G-x-G-x-x-G motif is marked as yellow. Hydrogen bonding interactions are indicated as black dashed lines. (**d**) Schematic interactions between NAD^+^ and A2HBD. Hydrogen bonding interactions are indicated as black dashed lines, whereas hydrophobic interactions are not indicated in lines.

**Figure 4 ijms-25-10711-f004:**
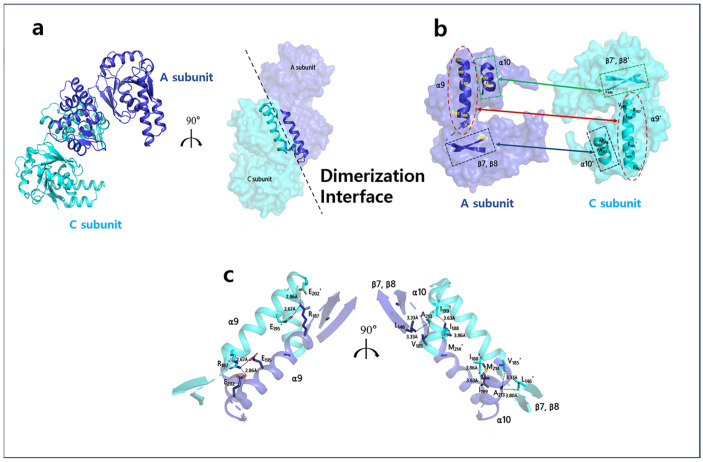
Dimerization interface of A2HBD (**a**) The representative snapshot of the crystal structure of the A2HBD dimer is shown in the cartoon (left) and the surface (right) model. In the cartoon diagram, the subunits A and C are denoted by blue and cyan colors, respectively. They are arranged in a “tail-to-tail” configuration, facilitated by interactions between their C-terminal domains. The surface diagram highlights only the secondary structures relevant to dimerization. (**b**) An “open-book” view of the dimerization interface between subunits A and C. The contact sites of the dimerization interface are indicated with red, green, and blue arrows; and dotted figures. (**c**) Core residues involved in dimerization interface. The combined ribbon and stick model depicts the electrostatic interactions (left) and hydrophobic interactions (right) between the secondary structure elements that contribute to the dimerization interface of the two subunits.

**Figure 5 ijms-25-10711-f005:**
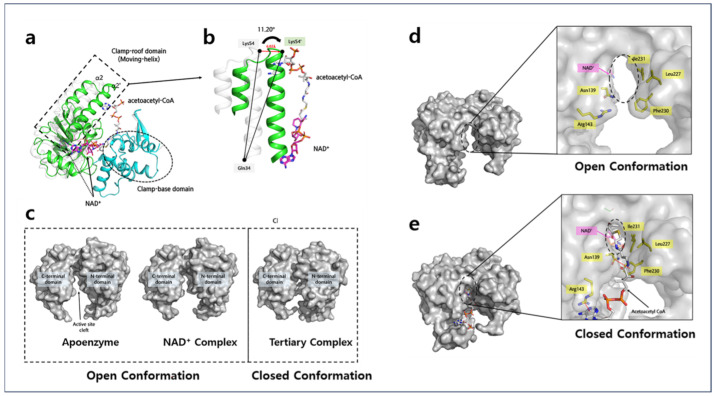
Domain shifting of A2HBD upon acetoacetyl-CoA binding. (**a**) Superimposition of NAD^+^ complex A2HBD (white) and tertiary complex (green and cyan). Clamp-roof domain movement (α2 to α2′) is described in (**a**) dotted box. NAD^+^ in cofactor complex, which is transparent, relocated to clear NAD^+^ (magenta) upon substrate binding. (**b**) The detailed cartoon diagram of clamp-roof domain shifting with (**a**) dihedral angle of 11 degrees upon substrate binding. (**c**) Surface diagram of open-to-closed conformation followed by substrate binding. (**d**,**e**) The open-close conformational change in residues of the catalytic cavity. Amino acid residues involved in forming catalytic cavities are shown as a stick model with a yellow color. “Catalytic cavity” is presented with a dotted circle. Upon substrate binding, the size of the cavity decreased by about 55%. The cation-π interaction (Asn139-Phe230) is shown as dotted red lines.

**Figure 6 ijms-25-10711-f006:**
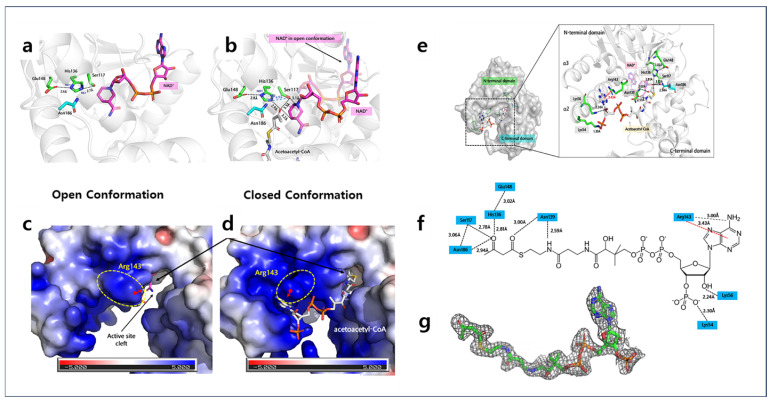
(**a**,**b**) The open-closed conformational change of interactions within catalytic residues in A2HBD upon substrate binding. In both panels, A2HBD structures are shown as gray-colored cartoons. The catalytic residues (Ser117, His136, Glu148) and NAD^+^ are shown as stick diagrams with green and magenta colors, respectively. (**a**) The open conformation occurs when only the NAD^+^ cofactor is bound. The hydrogen bonding interactions between catalytic residues are presented as dotted lines. Three catalytic residues are bridged with hydrogen bonding interactions. (**b**) Upon substrate binding, the ligands and the catalytic residues adopt closed conformation. The hydrogen bonding interactions are marked as dotted lines. The distance between the NE2 of His136 and the side chain oxygen of Ser117 residue is presented with a pale-blue-colored dotted line labeled as 4.9Å. The distance became more distant from 2.7Å to 4.9Å in the transition to closed conformation, indicating breakage of hydrogen bonding. (**c**,**d**) Surface electrostatic presentation of the substrate binding site; A2HBD structures without (**c**) and with (**d**) substrate binding are presented as surface electrostatic presentations with the same orientation. The arginine 143 residue is marked as a yellow dotted circle and labeled. The black arrow indicates substrate binding pocket. The orientation of the red-colored arrow indicates the downside of the guanidinium side chain of arginine. Following substrate binding, the guanidinium side chain rotates approximately 90 degrees to align parallel to the adenine group of acetoacetyl-CoA to stabilize the substrate binding. (**e**–**g**) Tertiary crystal structure of A2HBD in complex with NAD^+^ and acetoacetyl-CoA. (**e**) Overall tertiary complex of A2HBD. A2HBD is shown as a gray surface model. The acetoacetyl-CoA and substrate-interacting residues are shown as stick diagrams. (**f**) The acetoacetyl-CoA is presented as white and N-terminal residues and C-terminal residues interacting with the substrate. Hydrogen bonding interactions are presented as dotted lines and cation-π interaction is represented as red-colored dotted lines. (**g**) The 2fofc electron density map (1.0 σ) of acetoacetyl-CoA by SFX-data.

**Figure 7 ijms-25-10711-f007:**
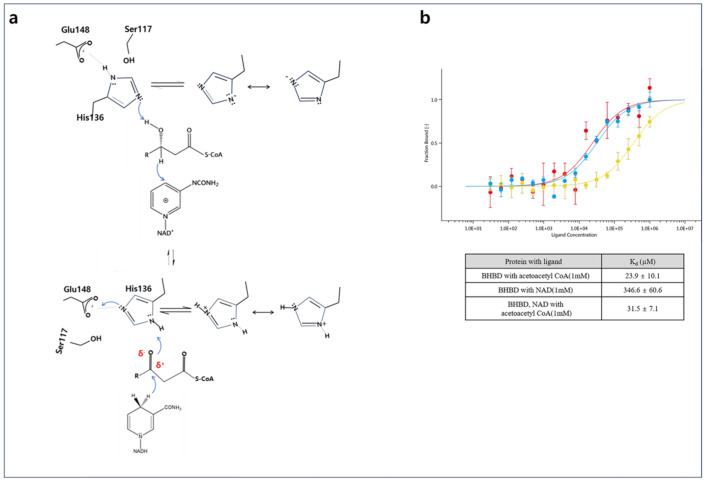
(**a**) Structure-based catalytic mechanism of the acetoacetyl-CoA-catalyzed reaction. Schematic diagram of proposed catalytic mechanism in A2HBD active site. Hydride transfer and nucleophilic substitution are presented as blue-colored arrows. Polarized carbonyl carbon and oxygen are indicated as red-colored δ^+^ and δ^−,^, respectively. (**b**) Binding affinity analysis of NAD^+^, acetoacetyl-CoA, and NAD^+^/acetoacetyl-CoA by MST experiments. The K_d_ values were showed BHBD with acetoacetyl-CoA in red color; BHBD in NAD^+^, and acetoacetyl-CoA in cyan color; BHBD with NAD^+^ in yellow color, respectively. Each K_d_ value is shown in the table.

**Table 1 ijms-25-10711-t001:** Data collection and refinement statistics for apo- and complex structures of 3-Hydroxybutyryl-CoA dehydrogenase.

	Apo	Native with NAD	S117A with AACoA	S117A with AACoA&NAD
**Data collection**				
X-ray source	PAL11C	PAL11C	SFX	PAL11C
Wavelength (Å)	1.000	1.000	1.000	1.000
Space group	P2_1_	P2_1_	P3_2_12	P3_2_12
a, b, c (Å)	87.66, 126.61, 108.09	90.82, 80.48, 128.35	91.01, 91.01, 212.7	90.56, 90.56,212.50
α, β, γ (°)	90.0, 110.2, 90.0	90.0, 102.2, 90.0	90, 90, 120	90, 90, 120
Resolution (Å)	48.04–2.69 (2.72–2.69)	46.64–2.55 (2.58–2.55)	45.51–1.90 (1.97–1.90)	45.28–2.20 (2.23–2.2)
Unique reflections	60,712 (1807)	54,530 (1522)	190,068 (7874)	50,509 (1392)
Completeness	96.7 (78.3)	91.7 (77.2)	99.99 (100.0)	99.0 (83.6)
Redundancy	6.8 (5.5)	4.1 (2.3)	3794 (189.7)	10.2 (3.8)
I/σ	10	23.1	26.1	15.6
Rmerge (%)	0.22 (0.75)	0.08 (0.25)	0.09 (0.26)	0.19 (0.70)
**Refinement statistics**				
Resolution (A)	48.04–2.69	46.64–2.55	45.51–1.9	45.28–2.20
Reflections used in refinement	60,712	54,530	190,068	50,509
Rwork/Rfree	0.2458/0.2943	0.2058/0.2661	0.1795/0.2224	0.1988/0.2442
**R.M.S. deviations**				
Bond lengths (A)	0.011	0.017	0.010	0.022
Bond angles (°)	1.18	1.24	1.26	1.4
**Ramachandran plot (%)**				
Favored	92.28	93.94	96.26	99.06
Allowed	6.16	5.65	3.51	0.94
**No. atoms**				
Protein	12,930	12,939	6423	6423
Ligands	0	264	98	98
Solvent	0	79	250	250
**B-factors**	52	62.3	26.2	37.4

Statistics for the highest resolution shell are shown in parentheses.

## Data Availability

The accession numbers for the structures of *Apo-A2HBD, A2HBD complex with NAD +, A2HBD (S117A) complex with acetoacetyl-CoA*, and *A2HBD (S117A) complex with NAD +/acetoacetyl-CoA* as described in this paper, are PDB9JIO, 9JHE, 9JHY, and 9JHZ, respectively. All other data are included in this paper.

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
