# Peer review of "Structural Insights and Catalytic Mechanism of 3-Hydroxybutyryl-CoA Dehydrogenase from Faecalibacterium Prausnitzii A2-165"

_ijms, 2024, doi:10.3390/ijms251910711_

Round 1

Reviewer 1 Report

Comments and Suggestions for Authors

In this study, the authors determined four crystal structures of 3-hydroxybutyryl CoA dehydrogenase from Faecalibacterium (apo form, NAD+ binding structure, substrate acetoacetyl-CoA binding structure, and NAD+ and acetoacetyl-CoA binding structure). Based on the three-dimensional structures, the binding mode of the coenzyme and substrate was clarified.

Major comments:

In the introduction, the authors states that 3-hydroxybutyryl CoA dehydrogenase from Faecalibacterium is the key enzyme involved in atopic dermatitis. However, the results of this study describe the three dimensional structures of 3-hydroxybutyryl CoA from Faecalibacterium. It does not describe how the study of the structure and reaction mechanism of this enzyme relates to the treatment of atopic dermatitis.

There are multiple publications on the three dimensional structures and reaction mechanism of hydroxybutyryl CoA dehydrogenases. It is necessary to refer the literatures appropriately to separate what is previously known about the structure and reaction mechanism from what is newly discovered in this study. As it is, it is very difficult to understand what has been revealed for the first time, and novelty cannot be judged.

Other points:

L116-117, “however, during the… cooperative process”: The meaning of the sentence is unclear.

L543-544, “There is a clear connection”: It is necessary to indicate specifically what clear connection is referring to. It is unclear.

Table 1: Some values are missing and the table is incomplete.

Author Response

Reviewer1

In this study, the authors determined four crystal structures of 3-hydroxybutyryl CoA dehydrogenase from Faecalibacterium (apo form, NAD+ binding structure, substrate acetoacetyl-CoA binding structure, and NAD+ and acetoacetyl-CoA binding structure). Based on the three-dimensional structures, the binding mode of the coenzyme and substrate was clarified.

Major comments:

  • In the introduction, the authors states that 3-hydroxybutyryl CoA dehydrogenase from Faecalibacterium is the key enzyme involved in atopic dermatitis. However, the results of this study describe the three dimensional structures of 3-hydroxybutyryl CoA from Faecalibacterium. It does not describe how the study of the structure and reaction mechanism of this enzyme relates to the treatment of atopic dermatitis.

==> Yes, we agreed. Regarding the reviewer’s comments, we changed this part in the introduction.

“Consequently, comprehending the function of key enzymes such as thiolase, 3-hydroxybutyryl-CoA dehydrogenase, butyryl-CoA dehydrogenase, and butyryl-CoA:acetate CoA transferase in the butyrate pathway is highly significant. Although the structures of these enzymes contributing to the production of various butyrates have been reported [12, 13], there have been no comparative studies of a difference in the production despite the same subspecies. To understand the differences in butyrate production between the L2-6 and A2-165 subspecies, we will aim to clarify the architectures of key enzymes from both subspecies. The A2HBD has elevated expression in F. prausnitzii A2-165 and is crucial for the transformation of acetoacetyl-CoA (Figure S1a) to 3-hydroxybutyryl-CoA in the butyrate cycle. We describe the whole structure of A2HBD, including the substrate and co-factor binding modes, as derived from three crystal structures (Figure 1a, 2a, 2b). Additionally, informed by these structural findings, we propose a putative molecular mechanism for the A2HBD enzyme. In the tertiary-A2HBD structure, the electron density of acetoacetyl-CoA and NAD+ were observed in only one protomer. We propose that this enzyme exhibits cooperativity upon substrate binding, resulting in a conformational change of the cleft. Furthermore, our biochemical findings demonstrated that the binding affinity of acetoacetyl-CoA is less stable than that of NAD+.”

==> We added this sentence in the conclusion.  “We will clarify the structures of HBD from F. prausnitzii L2-6 (L2HBD) and then compare the structures of A2HBD and L2HBD in a separate article.”

Other points:

  • L116-117, “however, during the… cooperative process”: The meaning of the sentence is unclear.

è Yes, we deleted it and revised it. “); In the tertiary-A2HBD structure, the electron density of acetoacetyl-CoA and NAD+ were observed in only one protomer. The exact reason for the complex formation involving only one subunit remains unclear. We propose that, upon substrate binding, the cleft formed between the A, C, and B, B' subunits contracts by approximately 11 degrees (Figure 2b, 2c).”

  • L543-544, “There is a clear connection”: It is necessary to indicate specifically what clear connection is referring to. It is unclear.

==> We revised the sentences in the conclusions.

“ To understand the butyrate metabolism pathway, we have determined the four crystal structures of 3-hydroxybutyryl-CoA dehydrogenase from Faecalibacterium prausnitzii strain A2-165 (A2HBD) in the presence of acetoacetyl-CoA and NAD+. This was achieved using the PAL-11C beamline (a 3rd generation facility) and PAL-XFEL (a 4th generation facility) with the technique of Serial Femtosecond Crystallography (SFX). In the tertiary-A2HBD structure, the electron density of acetoacetyl-CoA and NAD+ were observed in only one protomer. We propose that this enzyme exhibits cooperativity upon substrate binding, resulting in a conformational change of the cleft. Furthermore, our biochemical findings demonstrated that the binding affinity of acetoacetyl-CoA is less stable than that of NAD+. The Arg143 has a critical role in identifying substrates and facilitating catalytic activity. The binding mechanism of acetoacetyl-CoA was studied further, revealing that it is less stable than NAD+. This work validates the conformational changes of catalytic triads involved in the catalytic reaction. Furthermore, it presents a theorized mechanism for substrate reduction based on structural discoveries. Our biochemical findings demonstrated that the binding affinity of acetoacetyl-CoA is less stable than that of NAD+. We will clarify the structures of HBD from F. prausnitzii L2-6 (L2HBD) and then compare the structures of A2HBD and L2HBD in a separate article.”

  • Table 1: Some values are missing and the table is incomplete.

==> Yes, we revised it. Thanks a lot.

Reviewer 2 Report

Comments and Suggestions for Authors

This review concerns the article type manuscript entitled “Structural Insights and Catalytic Mechanism of 3-Hydroxybutyryl-CoA Dehydrogenase from Faecalibacterium prausnitzii A2-165”, submitted to International Journal of Molecular Sciences (Manuscript ID: ijms-3230414).

The article is consistent with the title.

The subject of the article corresponds to the aim of the IJMS journal.

The scientific achievement of the work is X-ray structure determination of four (4) proteins. The data for these proteins were deposited in Protein Data Bank as 9JIO, 9JHE, 9JHY, and 9JHZ (Hepatitis E virus capsid protein E2s domain (genotype I) in complex with Fab H4, 3-hydroxybutyryl-CoA dehydrogenase, 3-Hydroxybutyryl-CoA dehydrogenase mutant (S117A) with acetoacetyl CoA, 3-Hydroxybutyryl-CoA dehydrogenase mutant(S117A) with acetoacetyl CoA and NAD) with the status codes HPUB and HOLD (processing complete, entry on hold until publication and hold until a certain date).

In my opinion, this is enough to consider the work for publication.

Also, the work is clearly written, introduction gives the proper insight in the aim of the work, the results focus on details of the structure determination, references are adequate.

Nevertheless, some remarks should be considered in the revised version.

1.      Generally, the pictures in the figures are too small to be informative and does not help much in analysis of the text. My proposal is to increase the dimension of the pictures. If this is not acceptable (for example, due to limit of the pages of the manuscript), some of them should be shifted to Supplementary Materials and only necessary should be maintained in the main text.

2.      The Figure 6b should be corrected. The quality of drawing should be increased. The Lewis structure of the imidazole ring in non-protonated form is incorrect, because the pi nitrogen atom has four bonds and should have positive charge, whereas the tau nitrogen atom has only two bonds and should have negative charge. Such resonance structure with charge separation is highly unlikely. Instead, the double bond should be drawn to the tau nitrogen. Also, the number of arrows describing the shift of electrons should be the same at both sides of the equation. The arrows should start from electrons. Maybe this is detailed remark, however, it is difficult to explain students why in the professional scientific texts the schemes are not drawn properly.

3.      Details. Some part of the text is bold (see Figures 2, 3, and 5 legends, line 265). Spaces between the values and units (line 277, 382, 383, 405, 409). Figure 8 (line 416).

Author Response

Reviewer 2

This review concerns the article type manuscript entitled “Structural Insights and Catalytic Mechanism of 3-Hydroxybutyryl-CoA Dehydrogenase from Faecalibacterium prausnitzii A2-165”, submitted to International Journal of Molecular Sciences (Manuscript ID: ijms-3230414).

The article is consistent with the title.

The subject of the article corresponds to the aim of the IJMS journal.

The scientific achievement of the work is X-ray structure determination of four (4) proteins. The data for these proteins were deposited in Protein Data Bank as 9JIO, 9JHE, 9JHY, and 9JHZ (Hepatitis E virus capsid protein E2s domain (genotype I) in complex with Fab H4, 3-hydroxybutyryl-CoA dehydrogenase, 3-Hydroxybutyryl-CoA dehydrogenase mutant (S117A) with acetoacetyl CoA, 3-Hydroxybutyryl-CoA dehydrogenase mutant(S117A) with acetoacetyl CoA and NAD) with the status codes HPUB and HOLD (processing complete, entry on hold until publication and hold until a certain date).

In my opinion, this is enough to consider the work for publication.

Also, the work is clearly written, introduction gives the proper insight in the aim of the work, the results focus on details of the structure determination, references are adequate.

Nevertheless, some remarks should be considered in the revised version.

  1. Generally, the pictures in the figures are too small to be informative and does not help much in analysis of the text. My proposal is to increase the dimension of the pictures. If this is not acceptable (for example, due to limit of the pages of the manuscript), some of them should be shifted to Supplementary Materials and only necessary should be maintained in the main text.

==> Thank you for your insightful remarks. We redrew all figures, increasing their dimensions. Some figures were relocated to supplemental resources. Please see the manuscript.

  1. The Figure 6b should be corrected. The quality of drawing should be increased. The Lewis structure of the imidazole ring in non-protonated form is incorrect, because the pi nitrogen atom has four bonds and should have positive charge, whereas the tau nitrogen atom has only two bonds and should have negative charge. Such resonance structure with charge separation is highly unlikely. Instead, the double bond should be drawn to the tau nitrogen. Also, the number of arrows describing the shift of electrons should be the same at both sides of the equation. The arrows should start from electrons. Maybe this is detailed remark, however, it is difficult to explain students why in the professional scientific texts the schemes are not drawn properly.

==> We appreciate your kind comments. We redrawn the Figure 6b and shifted Figure 6a to Supplementary Materials. Please see Figure 7a.

  1. Some part of the text is bold (see Figures 2, 3, and 5 legends, line 265). Spaces between the values and units (line 277, 382, 383, 405, 409). Figure 8 (line 416).

==> Thanks a lot. We revised them.

Round 2

Reviewer 1 Report

Comments and Suggestions for Authors

The authors answer the reviewer's second question as follows. I do not judge this to be an adequate answer at all.

We added this sentence in the conclusion.  “We will clarify the structures of HBD from F. prausnitzii L2-6 (L2HBD) and then compare the structures of A2HBD and L2HBD in a separate article.”

Comparison with previous reports and results are needed for discussion.

If the authors do not indicate in this paper, I have no idea what is a new finding of this study. As it is, it is just a report of the three dimensional structure.

Author Response

Reviewer1

In this study, the authors determined four crystal structures of 3-hydroxybutyryl CoA dehydrogenase from Faecalibacterium (apo form, NAD+ binding structure, substrate acetoacetyl-CoA binding structure, and NAD+ and acetoacetyl-CoA binding structure). Based on the three-dimensional structures, the binding mode of the coenzyme and substrate was clarified.

“We added this sentence in the conclusion.  “We will clarify the structures of HBD

 from F. prausnitzii L2-6 (L2HBD) and then compare the structures of A2HBD and

L2HBD in a separate article.”

Comparison with previous reports and results are needed for discussion.

If the authors do not indicate in this paper, I have no idea what is a new finding of this study. As it is, it is just a report of the three dimensional structure.

==> Basically, we agree with you, but I think the main findings are in this paper nonetheless.

Our main findings are as follows.

  • We solved four new crystal structures from prausnitzii using 3rd and 4th generation synchrotron radiation.
  • We solved the tertiary structures complexed with cofactor and substrate. They exist hexamer in solutions. These tertiary complexes are found in one protomer. We may suggest the enzyme exhibits cooperativity upon substrate binding, resulting in a conformational change. We discuss them.
  • We measured the binding affinity of NAD, substrate or both by using MST. This value is shown to the relative stability of binding.
  • We analyzed the surface model or conformational change by comparing each complex structure.

Regarding your comments, therefore, we deleted “ We will clarify the structures of HBD from F. prausnitzii L2-6 (L2HBD) and then compare the structures of A2HBD and L2HBD in a separate article” in conclusion due to ongoing results. Regrettably, it will be challenging to incorporate the findings into this paper. I apologize for not providing you with the results; however, I trust that you comprehend the situation.

Round 3

Reviewer 1 Report

Comments and Suggestions for Authors

The manuscript has been well revised according to the reviewers’ suggestions. I recommend this manuscript for publication.